# The Lesser Chestnut Weevil (*Curculio sayi*): Damage and Management with Biological Control Using Entomopathogenic Fungi and Entomopathogenic Nematodes

**DOI:** 10.3390/insects13121097

**Published:** 2022-11-29

**Authors:** Camila C. Filgueiras, Denis S. Willett

**Affiliations:** 1Department of Biology, University of North Carolina Asheville, Asheville, NC 28804, USA; 2North Carolina Institute for Climate Studies, North Carolina State University, 151 Patton Avenue, Asheville, NC 28801, USA

**Keywords:** integrated pest management, plant-insect-microbe interactions, nut production

## Abstract

**Simple Summary:**

The lesser chestnut weevil (*Curcilio sayi*) is an emergent pest of chestnuts in the United States that can cause multifaceted damage and has limited management options. We explored the damage caused by *C. sayi* in a commercial chestnut orchard. Additionally, we evaluated potential management options for biological control. We found that *C. sayi* emerged from infested chestnuts more than four weeks post harvest and, in some cases, single nuts can host more than 10 *C. sayi* larvae. We also found that nut weight continues to decline even after *C. sayi* larvae have emerged from the chestnuts. Specific strains of entomopathogenic nematodes increase the mortality of *C. sayi* larvae. Biological control using entomopathogenic fungi and entomopathogenic nematodes could be complementary approaches to managing this pest and reduce *C. sayi* populations and chance of damage.

**Abstract:**

The lesser chestnut weevil, *Curculio sayi* (Gyllenhal), can cause irreparable damage to chestnuts through direct consumption and/or introduction of secondary pathogens. With the resurgence of blight resistant American Chestnut plantings both for commercial production and for habitat restoration, *C. sayi* has become a similarly resurgence pest. Here, we investigated the nature and extent of *C. sayi* larval damage on individual nuts and collected harvests with an eye toward the quantifying impacts. Next, we explored management options using biological control including entomopathogenic fungi and entomopathogenic nematodes. Nut damage from *C. sayi* can be extensive with individual nuts hosting several larvae, larvae emerging from nuts several weeks post harvest, and nut weight loss even after *C. sayi* have emerged from the nut. Applications of entomopathogenic fungi reduced chances of chestnut infestation, while certain strains of entomopathogenic nematodes increased the probability of *C. sayi* larval mortality. Understanding *C. sayi* damage and exploring biological control management options could be a useful tool in the effective management of this resurgent pest.

## 1. Introduction

American Chestnut (*Castanea dentata*) was once the foundation species for forests throughout Appalachia along the east coast of the United States. Occupying at least 50% of the basal area across 800,000 square kilometers, American Chestnut trees were once estimated to have numbered more than 4 billion in the early 1900s [1]. These trees produced a prodigious amount of chestnuts. In 1911, a single West Virginia train station shipped 155,000 lbs of chestnuts with a retail market value of nearly $750,000 USD today [2]. These chestnuts were a staple of animal and human diets; nutritious chestnuts were used to fatten livestock before the market and were an ubiquitous commodity in early American life [2,3,4]

With the introduction of chestnut blight (*Cryphonectria parasitica*) in the early 1900s, American Chestnut trees suffered a precipitous and rapid decline. By 1960, American Chestnut trees were almost completely absent from the canopies of eastern forests [1]. It was only in the past decades that blight resistant hybrids made the resurgence of chestnut production possible. The development of these blight resistant hybrids have opened opportunities for commercial expansion of chestnut as a high value specialty crop. Although blight resistant hybrids have largely overcome the challenges imposed by chestnut blight, commercial resurgence of American Chestnut Hybrids in recent years has been accompanied by the resurgence of another organism: the lesser chestnut weevil (*Curculio sayi*).

The lesser chestnut weevil was documented extensively in the early 1900s as being a prominent pest of chestnuts. Reports mentioned that large losses in chestnut production occurred due to infestation by *C. sayi* larvae [5]. Infestation by *C. sayi* was recognized to range widely with rates between 50–75% considered normal [6]. Infestation rates as high as 100% were also reported [6]. The extent of these pest problems made it a public safety concern with federal and state governments seizing large quantities of chestnuts because they contained large amounts of *C. sayi* larvae and excreta [6].

While *C. sayi* numbers decreased alongside its host, the American Chestnut, this weevil is once again becoming a prominent pest of commercial chestnut plantings, rapidly emerging in as little as two years to high levels of infestation [7]. With bivoltine populations in the southern United States and univoltine populations in the north, we now realize that *C. sayi* damage takes two primary forms [7].

The first is physical damage. Consumer confidence in chestnuts takes an incredible blow when larvae emerge on the counter at home. Even if the larvae are killed in post-harvest heat treatments, excessive damage can also substantially reduce nut-meat. The second is damage through facilitation of infections. *C. sayi* larvae are associated with *Aspergillus* fungi [8] which produce the diarrheagenic toxin emodin [9].

Despite these negative impacts, options for control are limited. Few commercial products are labeled for use against *C. sayi* and those commercial pesticides that are used have potentially toxic environmental effects [10]. Post harvest control methods for treating larvae in collected chestnuts abound but, even if they are effective, they only treat the symptoms. Growers still lose production and it is unclear whether post-harvest treatments can effectively address fungi. Biological control has been implemented to control other related *Curculio* weevils, but their efficacy with *C. sayi* is unknown [11,12,13,14,15].

To explore options for biological control, entomopathogenic fungi, entomopathogenic nematodes, and their combination were evaluated in commercial chestnuts over two years. Because emerging *C. sayi* adults tend to climb to the chestnut canopy in the spring, we thought that topical trunk applications of entomopathogenic fungi could be effective at targeting the adult lifestage. Because *C. sayi* larvae burrow into the soil to pupate and overwinter, we thought that entomopathogenic nematodes could be effective at targeting the larval life stage. Both these biological control options, alone and combined, have been used to control *Curculio elephas* but never used against *C. sayi* [12,14,15,16,17]. In addition, damage from *C. sayi* larvae was quantified to address and document anecdotal reports of damage that varied widely.

## 2. Materials and Methods

To evaluate the damage and management of *C. sayi*, we used a commercial chestnut stand at the Rose Valley Farm (43°09026.500 N, 76°55021.100 W) in upstate New York. This commercial stand was composed of a blend of mature (15+ years old) American Chestnut Hybrids planted approximately 7 m on center and forming a complete canopy after leafing out. Catkins (flowering) occurred in mid to late June and nut drop began in late September to early October. The soil type was Elnora Loamy Fine Sand.

### 2.1. Chestnut Weevil Damage

To evaluate the damage caused by *C. sayi*, chestnuts were collected within 24 h of nut drop and brought to the lab in two different cohorts.

For the first cohort, fifty chestnuts from each of four treatments (for a total of 200 chestnuts, treatments described below) were placed individually into 50 mL Falcon Tubes. Each of these nuts were evaluated every 24 h and the number of larvae that had emerged from the nuts counted.

For the second cohort, chestnuts sufficient to fill a 1 L plastic planting container were collected from the control treatment (described below). These planting containers were perforated on the bottom to allow for larval egress after emergence from the chestnuts. Ten replicates of these containers were collected. Every 24 h for 45 days these containers were evaluated. The number of larvae that had emerged were counted and weighed. The total weight of the nuts in the container were also weighed.

### 2.2. Entomopathogenic Nematode Strain Evaluation under Lab Conditions

To evaluate the ability of entomopathogenic nematode strains to control *C. sayi* larvae, we evaluated the mortality of late instar *C. sayi* larvae exposed to thirteen different strains of entomopathogenic nematodes collected from the east coast of the United States. Late instar *C. sayi* larvae were collected as they emerged from harvested chestnuts and placed individually into wells of ELISA plates lined with 1 cm diameter filter paper (Whatman #1(St. Louis, MO, USA)). Each of these wells received 200 µL of liquid; controls received water only while treatments received a nematode solution containing 1000 infective juveniles of the appropriate strain per mL for approximately 200 nematodes per larvae. Treatments were replicated 24 times and controls replicated 96 times across ELISA plates. *C. sayi* mortality was monitored for 10 days.

### 2.3. Chestnut Weevil Management

To evaluate the viability of biological control techniques for the management of *C. sayi*, we evaluated both the application of entomopathogenic nematodes (EPN) to the soil and application of entomopathogenic fungi (EPF) to the trunks of chestnut trees. To do so, we utilized a two factor, full factorial design with four treatments: EPN Only, EPF Only, Both EPN and EPF, and the Control (neither EPN nor EPF). We utilized a randomized controlled block design and treated individual monitoring locations (approximately four trees) as replicates.

#### 2.3.1. Chestnut Weevil Biological Control Applications

Entomopathogenic nematodes were applied to the soil using a 15L backpack sprayer (Chapin 61800, Chapin Manufacturing, Batavia, NY, USA) at a rate of 120 million per acre. Two complementary species of entomopathogenic nematode were used: *Steinernema feltiae* and *Heterorhabditis bacteriophora* both collected from upstate NY and reared in *Galleria mellonella* larvae following previously established methods [18,19]. These species were chosen for their established hardiness in similar environments. Infective juveniles (IJs) of both species were collected after emergence from their hosts and used within seven days. The two species were mixed at a 1:1 ratio to a concentration of 2000 IJs/ml. Prior to each application, samples were taken in the field from the backpack sprayer before and after passing through the sprayer to ensure the viability for application. EPNs were applied three times each field season on May 23rd, June 6th, and July 1st in both 2019 and 2020. Plots not receiving the nematode treatment instead received the same amount of water delivered via a separate backpack sprayer, but without nematodes.

Entomopathogenic fungi were applied to the trunks of each tree in the treatment in a 0.5 m band entirely circling the circumference of the tree, terminating approximately 1 m above ground level using a large paintbrush similar to whitewashing. *Beauveria bassiana*, commercially available in Mycotrol ESO (Bioworks, Victor, NY, USA) was used as the entomopathogenic fungi at the recommended rate (1 quart/acre) on June 17th in both 2019 and 2020. The efficacy of this product was confirmed by sampling the solution immediately prior to the field application then subsequent application to *G. mellonella* larvae in the lab. Plots not receiving the EPF treatment instead received the same amount of the solution but without *B. bassiana*.

The efficacy of these treatments was evaluated using pyramid and trunk traps to monitor adult *C. sayi* populations. Pyramid traps (Tedders Pyramid Trap, GL-5000-06, Great Lakes IPM, Vestaburg, MI, USA) are designed to attract newly emerging *C. sayi* adults as they climb adjacent vertical objects. These traps were staked to the ground approximately 0.5 m from adjacent tree trunks. Trunk Traps (Circle Trunk Trap, Small GL-4000-06, Great Lakes IPM, Vestaburg, MI, USA) were affixed to the tree trunks at breast height (1.35 m) such that the mesh screen of the trap directly abutted the trunk of the tree. These traps are designed to arrest and *C. sayi* adults ascending the trunks of the trees. These traps were emplaced in late April before the start of each season and monitored biweekly.

#### 2.3.2. Microcosms

Microcosms consisted of 20, five gallon (18.93L) plastic buckets into which 15–20 large diameter (about 7cm) holes were drilled at the sides and bottom of the bucket and then covered with a fine mesh screen (1000 micron). These microcosms were placed adjacent to the commercial chestnut orchard by excavating 20 individual holes for each microcosm, placing the buckets in the holes, then backfilling the soil into each bucket. The soil was allowed to resettle for one week then 25 *C. sayi* late instar larvae (that had emerged from collected chestnuts within 48 h) were placed into each microcosm and sealed inside with a mesh cap on the buried buckets. This arrangement ensured that the *C. sayi* could not escape the microcosms but that other microorganisms and water could move in and out via the soil.

Each of the four treatments described above were replicated five times in a randomized controlled block design. Each microcosm in each treatment received 500 mL of solution. Control microcosms received 500 mL of water. Entomopathogenic nematode-only treatments received a 500 mL of a solution containing *S. feltiae* and *H. bacteriophora* in a 1:1 ratio at 1000 IJs/ml for each species (2000 IJs/ml total). Entomopathogenic fungi only treatments received 500 mL of Mycotrol ESO *B. bassiana* solution (made by mixing 3.75 mL of Mycotrol ESO in water for a 500 mL total volume). Treatments receiving both entomopathogenic nematodes and entomopathogenic fungi received the same amount of total volume with both EPN and EPF treatments mixed. The application of treatments occurred one week following introduction of the *C. sayi* larvae.

Microcosms were initially emplaced in late October 2019 with the introduction of *C. sayi* larvae and application of treatments accomplished by late November 2019. Microcosms remained unperturbed until November 2020 when they were removed, taken to the lab, and carefully excavated. As a first step in the excavation, the covering screen was removed and the surface examined for emerged *C. sayi* adults. Following this initial examination, the soil was carefully excavated and delicately passed through a fine mesh seive to extract any remaining *C. sayi* adults, pupae, or larvae.

### 2.4. Analysis

Nut damage by emerging *C. sayi* larvae was visualized using histograms and density distributions then modeled using generalized additive models.

Mortality of *C. sayi* larvae resulting from nematode infection was evaluated using logistic regression considering the interaction between nematode strain and days post inoculation as factors. Best fit models were chosen after consideration of all potential interactions, residual analysis, goodness of fit metrics, and likelihood ratio tests. Post-hoc treatment comparisons were conducted using the Dunnett method of contrasting treatments with the control.

The efficacy of *C. sayi* management strategies was evaluated using poisson regression models for trap catch and emergence from microcosms. Logistic regression was used to evaluate nut infestation from respective treatments. Best fit models were chosen after consideration of all potential interactions, residual analysis, goodness of fit metrics, and likelihood ratio tests. Post-hoc treatment comparisons were conducted using custom contrasts for desired comparisons and the Bonferroni correction for adjusting the family-wise error rate.

All analysis was conducted in R version 4.2.1 [20] using RStudio [21] as an integrated development environment (IDE). The following packages facilitated the analysis, modeling, and visualization of results: tidyverse for data preparation and plotting [22]; car and emmeans for model evaluation and post-hoc comparisons [23,24]; ggpubr, here, and cowplot for assistance in plotting and figure generation [25,26,27].

## 3. Results

### 3.1. Chestnut Weevil Damage

Larval *C. sayi* emerged from individual chestnuts at varying rates and over an extended period of time. Out of the chestnuts with emergent larvae, eight had only one larva emerging in the time under observation (Figure 1A). A few chestnuts had more than five larvae emerging with three individual chestnuts hosting more than 10 *C. sayi* larvae that emerged under the period of observation. More than three quarters of chestnuts (75.8%) with larval emergence had more than one larva emerging over the observation period.

Larval *C. sayi* began emerging within a few days of the field collection of nuts dropped in the previous 24 h (Figure 1B). Median first emergence of larval *C. sayi* occurred between 10 and 15 days after harvest with median peak emergence occurring at 15 days. The median last observed emergence of *C. sayi* was at close to 30 days post harvest with the last *C. sayi* emerging at 45 days after harvest.

In examining cohorts of chestnuts collected in planting containers after dropping in the previous 24 h, the numbers of emerging larvae peaked at six days post harvest (Figure 2A). Following the peak, larval emergence tailed off with the last observed larval emergence occurring 31 days post harvest. Cumulative weight of the larvae emerging from collected chestnuts flatlined at 16 days post harvest (Figure 2B) and remained mostly flat for the remaining period of observation. The nut weight of collected nuts declined almost linearly over the period of observation (Figure 2C), even after 16 days post harvest.

### 3.2. Entomopathogenic Nematode Strain Evaluation under Lab Conditions

Entomopathogenic nematode strain, days post inoculation, and their interaction significantly explained the observed *C. sayi* mortality (Treatment: $\chi^{2}$ = 131.2, df = 13, p<0.001; Days: $\chi^{2}$ = 13.2, df = 1, *p* = 0.0003; Interaction: $\chi^{2}$ = 114.2, df = 13, p<0.001). Exposure to many nematode strains resulted in a significantly higher probability of mortality (Figure 3) compared to the controls.

### 3.3. Chestnut Weevil Management

Biological control treatments significantly explained the differences in *C. sayi* responses. Biological control treatments and year (but not their interaction) significantly explained adult *C. sayi* trap catch (Treatment: $\chi^{2}$ = 14.7, df = 3, *p* = 0.002; Year: $\chi^{2}$ = 208.9, df = 1, p<0.001). Entomopathogenic fungi (EPF) significantly reduced adult *C. sayi* trap catch as a main effect and compared to controls (*p* = 0.007, 0.03; Figure 4A). Adult *C. sayi* trap catch was significantly less (p<0.0001) in 2020 compared to 2019 (Figure 4B).

Biological control treatments significantly explained the probability of larval *C. sayi* emergence from chestnuts ($\chi^{2}$ = 11.96, df = 3, *p* = 0.008). Chestnuts from the treatment receiving only entomopathogenic fungi had a significantly lower chance of weevil infestation both as a main effect and compared to the control treatment (*p* = 0.006, 0.02; Figure 4C).

Biological control treatments also significantly explained the number of weevils recovered from microcosms ($\chi^{2}$ = 16.27, df = 3, *p* = 0.001). Treatments receiving entomopathogenic nematodes only had a significantly lower (*p* = 0.007) number of weevils recovered compared to the EPN + EPF treatment (Figure 4D). The average (±SD, SEM) recovery rate of weevils from microcosms across all treatments was 12% (±14%, 3%) or approximately 3 (±3.4, 0.75) weevils.

## 4. Discussion

### 4.1. Chestnut Weevil Damage

*Curculio sayi* larvae emergence exceeded expectations. While early reports of *C. sayi* emergence documented the ability for multiple larvae to emerge from a single nut [5,6,28], many modern anecdotes focused on the one larva, one nut rule of thumb. This is not the case. While some of the chestnuts with observed larval emergence did only have one larva, three quarters of the chestnuts with observed larval emergence had two or more larvae. One single chestnut contained 12 larvae.

This abundance of *C. sayi* larvae in a single nut could suggest that females do not have a mechanism for identifying chestnuts already with eggs. It is also unclear if there are advantages or disadvantages to having more than one larva in a single nut. There were not obvious differences in the health of the larvae emerging from the nuts with 10 or more larvae compared with the nuts containing a single larvae, although we did not accompany each larvae to adulthood.

Emergence of *C. sayi* was also distributed in time. While peak emergence (when most of the larvae in a nut emerged) tended to occur between 10 and 24 days post harvest, the emergence of the first weevil from an individual chestnut occurred as early as a few days following harvest and as late as 42 days post harvest. This long time period for larval emergence was corroborated in observations of planting containers of chestnuts with the last larva emerging 31 days post harvest.

The distributed nature of *C. sayi* emergence suggests that post harvest control strategies may need to be adapted. A practice that involves waiting for two weeks (14 days) to determine which nuts are not infested does not seem adequate. Even nuts with no emergence after 14 days may still contain active larvae that could emerge potentially even three weeks later.

Waiting six weeks (42 days) to evaluate larval emergence and identify healthy nuts is also not an ideal option. Irrespective of time to market considerations, nut weight appears to decline linearly with time. This is true even after most larvae have left the nuts. Peak emergence of *C. sayi* in buckets of chestnuts occurred at six days post harvest and dropped to low levels by day 16 when the cumulative weight of the larvae flatlined. Despite very little emergence occurring after day 16, nut weight continued its almost linear decline.

This decline suggests that other factors are at play in decreasing nut weight, not solely *C. sayi* consumption of nut meat. A potential culprit is a bacterial/fungal complex associated with *C. sayi* larval tunnels inside the nuts. Feeding by *C. sayi* larvae open channels in the chestnut, which are colonized by potentially harmful microorganisms such as *Aspergillus* fungi [8,9,29,30]. These microorganisms are present long after *C. sayi* larvae have emerged from the nut. Their presence could be a contributing factor to the continued decline of nut weight over time. Independent of the causes, the more time between harvest and market delivery of chestnuts, the less chestnut material is available.

### 4.2. Chestnut Weevil Management

Because of the losses caused by *C. sayi*, management techniques that address burgeoning populations are critical. Applications of entomopathogenic fungi to trunks of mature chestnut trees significantly reduced the adult *C. sayi* trap catch compared to the control.

This reduction in trap catch could potentially be attributed to two possible factors. First, the applications of entomopathogenic fungi to the trunks of chestnut trees could have potentially served as a repellent to adult *C. sayi* moving to the tree canopy. Second, applications of entomopathogenic fungi could have had an effect of *C. sayi* mortality. While not immediately effective in this regard, insects coming into contact with entomopathogenic fungi can be colonized by this fungus and die in a matter of days from a systemic fungal infection.

Irrespective of the mechanism, the reduction in adult *C. sayi* trap catch was also reflected in the chance of weevil infestation in collected chestnuts. Chestnuts collected from the EPF treatment had a lower chance of being infested with *C. sayi* larvae than other treatments. This suggests that the effect of application of entomopathogenic fungi, translates into a reduction in weevil infestation of harvested chestnuts.

Treatments receiving both entomopathogenic nematodes and entomopathogenic fungi did not respond the same as entomopathogenic fungi only treatments. While these treatments were comparably lower in terms of adult *C. sayi* trap catch and chance of infestation, in each case, treatments of entomopathogenic fungi lowered the respective response compared to the entomopathogenic nematode-only treatments.

The lack of response of the entomopathogenic nematode treatments in the field was surprising given that many entomopathogenic nematode strains significantly increased the probability of *C. sayi* larvae. Although the strains chosen in this study were selected more for their ability to survive northern winters as opposed to virulence alone, they still increased the probability of larval mortality in laboratory trials. In particular, in one of the species we used, (*S. feltiae*) can significantly increase the probability of mortality compared with controls. While it is entirely possible that applications of entomopathogenic nematodes had no effect on *C. sayi* populations, it is also possible that the effect might be lagged. The drastic drop in *C. sayi* populations between years might be the result of the efficacy of entomopathogenic nematodes, which are are highly effective mobile agents that seek out weevil larvae. A similar effect has been observed in applications of entomopathogenic nematodes to control the pecan weevil (*Curculio caryae*); repeated applications of entomopathogenic nematodes resulted in the improved control of *C. caryae* in year two [31].

Because chestnut weevils in general, and likely *C. sayi* as well, seem to demonstrate a staggered approach to cohort development with potentially multi-year periods in the soil as eclosed adults, timing in using entomopathogenic nematodes could be critical [7,32,33]. Entomopathogenic nematodes applied at the beginning of the season might be helpful in establishing populations of these nematodes in the soil, but at the beginning of the season, most *C. sayi* in the soil are in nematode-resistant pupal cells, as more nematode resistant adults [7,34].

In contrast, applications of entomopathogenic nematodes at the end of the season might be more effective. At the end of the season, *C. sayi* are emerging from chestnuts and burrowing into the soil as larvae prior to establishing protections from soil organisms like entomopathogenic nematodes. The efficacy of these applications would not appear, however, until the next season, when the impact of this mortality on the emerging cohort would become apparent. If entomopathogenic nematodes were effective, we would expect this to become apparent over the course of two to three years as we began to see in this study and as has been observed in other field settings [35,36]. In addition, our microcosm experiments also indicated that applications of entomopathogenic nematodes could reduce populations of *C. sayi* when introduced to the soil as larvae.

## 5. Conclusions

Together, entomopathogenic nematodes and entomopathogenic fungi could present an option for the biological control of *C. sayi* infestations in commercial chestnut orchards. This dual approach has been evaluated in other similar systems with the greater chestnut weevil (*C. elephas*) [14]. Applications of entomopathogenic fungi could reduce chestnut infestations initially, while applications of entomopathogenic nematodes could reduce *C. sayi* populations over the longer term. There might even be a synergistic effect to this approach that emerges over the course of three to five years. This is the focus of future work.

## Figures and Tables

**Figure 1 insects-13-01097-f001:**
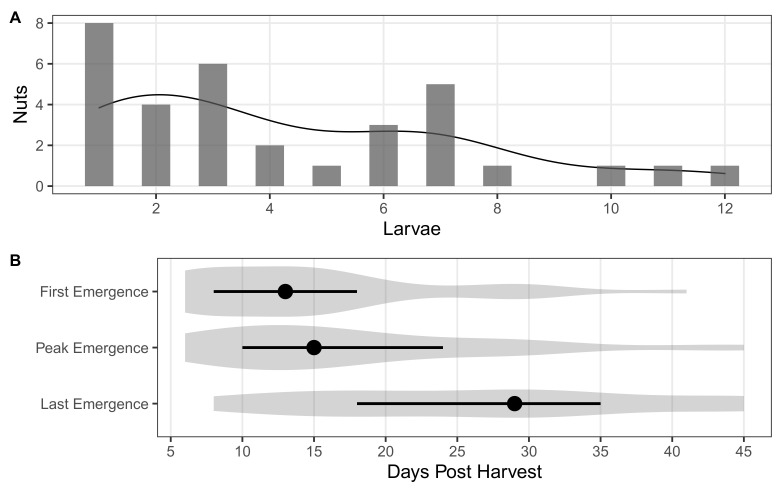
Larval *C. sayi* emergence from individual chestnuts. (**A**) Number of *C. sayi* larvae emerging from each individual chestnut. Each bar indicates the number of nuts having that many *C. sayi* larvae emerge. The line indicates the smoothed density distribution of that emergence. (**B**) Time to emergence by *C. sayi* larvae. First emergence denotes the amount of time to when the first *C. sayi* larva was observed leaving the nut. Peak emergence denotes the time to when the most number of larvae were observed leaving the nut on a given day. Last emergence denotes the time to when the last *C. sayi* larva was observed emerging from the chestnut. Shaded grey areas denote density distributions for each metric. Points denote the median. Error bars denote the first and third quartiles.

**Figure 2 insects-13-01097-f002:**
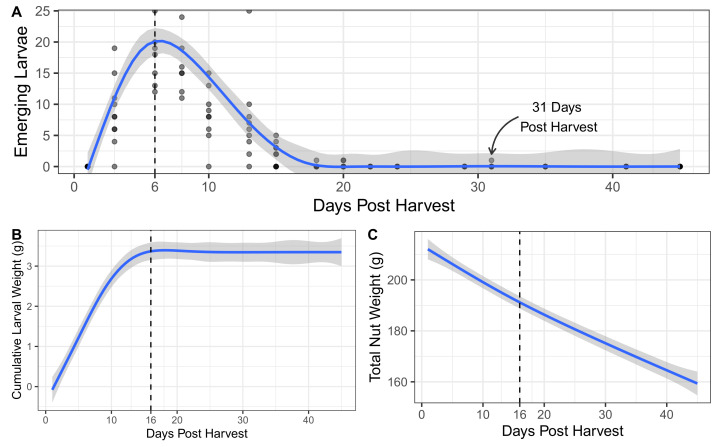
Larval *C. sayi* damage on groups of chestnuts. (**A**) Number of emerging larvae from 1 L planting containers of chestnuts. (**B**) Cumulative weight of all emerging larvae emerging from each 1 L planting container of chestnut. (**C**) Total weight of chestnuts in each 1 L planting container of chestnuts. For all plots: points denote individual observations; blue lines and shaded areas denote generalized additive model smoothed fit and 95% confidence intervals, respectively.

**Figure 3 insects-13-01097-f003:**
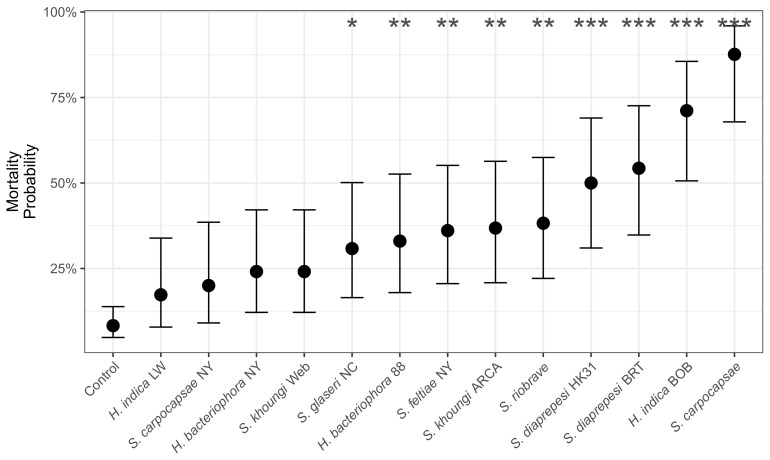
Probability of *C. sayi* mortality resulting from exposure to entomopathogenic nematode strains. *H. bacteriophora* NY and *S. feltiae* NY were the strains used in the field and microcosm trials. Mortality probability is calculated as of four days post inoculation. Points and errorbars denote mean and 95% confidence intervals, respectively. Asterisks denote significant differences at p<0.05 (*), p<0.01 (**), and p<0.001 (***) compared to controls, not among strains.

**Figure 4 insects-13-01097-f004:**
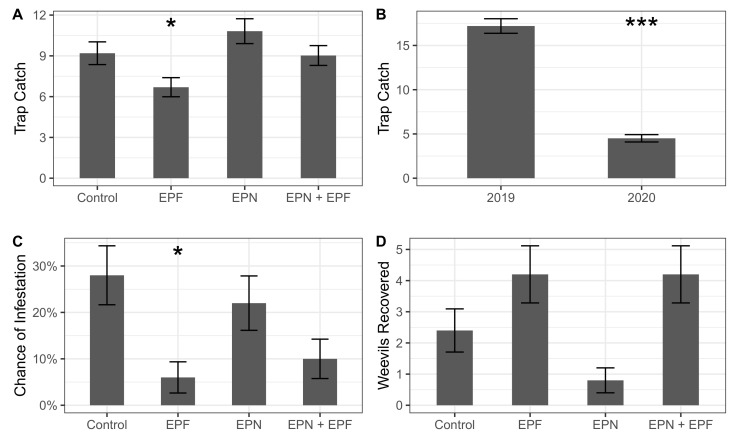
Biological control of *C. sayi*. (**A**) Effect of biological control treatments on *C. sayi* adult trap catch in a commercial chestnut orchard. Bars and error bars denote rate of trap catch in each treatment on a per trap basis and standard error, respectively. Single asterisk denotes a significant (p<0.05) difference from the control treatment. (**B**) *C. sayi* adult trap catch in each year of this study. Bars and error bars denote rate of trap catch in each year on a per trap basis and standard error, respectively. Multiple asterisks denote significant (p<0.001) difference from 2019. (**C**) Likelihood of a nut from a given treatment being infested by *C. sayi* larvae. Bars and error bars denote the chance of a nut from a given treatment being infested with *C. sayi* larvae and the standard error, respectively. Single asterisk denotes the significant (p<0.05) difference from the control treatment. (**D**) *C. sayi* recovered from microcosms by treatment. Bars and error bars denote weevils recovered by treatment and standard error, respectively. Single asterisk denotes significant (p<0.01) difference from EPN+EPF treatment and marginally significant (p=0.09) difference from the control.

## Data Availability

Data and code supporting this publication will be made available via GitHub following publication.

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
