# Peer review of "The Lesser Chestnut Weevil (Curculio sayi): Damage and Management with Biological Control Using Entomopathogenic Fungi and Entomopathogenic Nematodes"

_insects, 2022, doi:10.3390/insects13121097_

Round 1
Reviewer 1 Report
Article
The Lesser Chestnut Weevil: Damage and Management with Biological Control
Interesting article that addresses the effect of entomopathogenic nematodes and entomopathogenic fungi on the emergence of Curculio sayi. The introduction requires more work, it is somewhat confusing and does not flow well. The methodology might require a re-organization and better subheading titles. Result section is easy to follow, minor changes required. The discussion provides and discusses interesting management tools and when these tools should be employed. The discussion is relatively easy to follow.
The excessive use of quotes in the introduction worries me a little. This could be considered some sort of plagiarism even though the authors give proper credit. Quotes used to be a common tool not that long ago, however, time has changed. Nowadays, quotes are frowned upon, they can only be used for a universal definition. I would highly recommend paraphrasing these quotes to avoid misunderstanding.
Several statements need references.
There are several sentences that are wordy, therefore, are hard to read. I suggest the authors should delete words that don’t really add anything to the paragraph. Some suggestions are given in the comments.
Decision: Major revision required
Please find my comments below.
L2-4: Please consider “We explored the damage caused by C. sayi in a commercial chestnut orchard”. “Additionally, we evaluated potential management options for biological control”
L4: delete “can”; “emerged” instead of “emerge”
L5-6: “individual nuts can host more than 10 C. sayi larvae”
This might be an exception and should be written as one. Please consider: “and in some cases single nuts can host more than 10 C. sayi larvae”
L10: add author to the scientific name Curculio sayi
L11: please consider: “direct consumption and/or introduction of secondary plant pathogens”
L13: Delete “Management options can be limited, however”
L14: “investigated” instead of “investigate”
L15: add “,” after the word “next”; “explored” instead of “explore”
L19-20: “Biological control could be a promising method of control with potential effects developing over time”
This sentence is not clear and does not add anything to the paragraph (it’s vague). I suggest deleting it. However, if the authors wish to keep it, it should be rephrased. The next sentence is way better.
L21: in the effective management of this resurgent pest
L23: It might be common to the authors but not to other readers. Please add the scientific name of American Chestnut, order, and family.
L27: “in the early 1900s” instead of “around the turn of the 20th century”. Please avoid the use of unnecessary words.
L31-32: Please delete “that Thoreau wrote that “these nuts… were a good substitute for bread”.
Scientific publications should avoid copy/pasting phrases from other sources, even if these are cited. Avoid using quotes unless is absolutely necessary. Most current publication do not quote other sources, they paraphrase them.
L33: close parenthesis after the scientific name
L36-37: please consider “blight resistant hybrids made the resurgence of chestnut production possible”
L38-39: please consider “high value commodity” instead of “high value specialty crop”
L41: “of” instead of “in”
Since here is the first time the lesser chestnut weevil is named, please add the scientific name and you can delete it from the next sentence.
L42-43: “around the turn of the century” what century are the authors talking about. Please be specific.
L43-51: I have a great concern in this paragraph. Although all the quotes have been cited, they should not be used that way. This can easily be considered plagiarism and it’s against the journal policy. These quotes should be rephrased, paraphrased to avoid plagiarism. Please address accordingly.
L52-53: Please consider: “While C. sayi numbers decreased alongside its host the American Chestnut”
L57: delete “actual”
L59-60: Please consider “The second is introduction of secondary plant pathogens”
L62-64: “Few commercial products are labeled for use against C. sayi and those commercial pesticides that are used have potentially toxic environmental effects”. This sentence needs references.
L67-68: “Biological control has been implemented to control other related weevils” This sentence needs references.
L71-73: please consider: “Both these options, alone and combined, have been used to control Curculio elephas but never used against C. sayi”.
MATERIALS AND METHODS
This section needs to be re-organized. Please consider the following and titles order:
· Study site
· Chestnut weevil management (a more descriptive title would be better)
o Commercial orchard (This subheading needs a better description)
o Microcosms (This subheading needs a better description)
· Chestnut weevil damage (a more descriptive title would be better)
· Statistical Analysis
L75: You already stated the scientific name of the lesser chestnut weevil. The scientific name is no longer needed.
L78: “planted approximately 7m on center and forming” I do not understand this. Please clarify. What is 7m on center?
L105: “following previous established methods” This needs a reference. Where can I find this methodology?
L115: the correct name is Beauveria bassiana. Please address this in the document.
L116: Please state in the document what was the recommended rate for B. bassiana.
L130: “Microcosms”. This title needs a better description, please try to provide a better title.
L132: please delete the symbol about (7 cm); “holes were drilled” where? At the bottom? At the side? Please remember to provide enough details so your experiment can be replicated by others. The fine mesh screen was placed at the bottom of the bucket? What was the size of the mesh? Fine is just a vague term.
L152-155: Was this part processed in the lab? Or in the field? The way it’s written suggests that examination occurred in the field.
L156: Suggested title “Statistical analysis”
RESULTS
There are some missing details in this section. Please see comments below.
L173-179: Here the authors talk numbers of larvae emerging from individual chestnuts all together. However, in the methodology it clearly states that chestnuts were collected from 4 treatments. Why the authors pooled the data together? It would have been more interesting to know the number of larvae emerging from chestnuts associated with their respective treatments. Please address accordingly.
L174-175: “out of the chestnuts with emergent larvae, the eight had only one larva emerging in the time under observation”. I do not understand this statement, it is poorly written. The eight? What eight? The authors suggest looking at Figure 1. However, figure 1 is confusing as well. Why the dependent variable (number of larvae emerging) is in the X-axis? A graph should be easy to follow and interpret without looking at other sections in the document. Number of emerging larvae should be in the Y-axis (dependent variable), and the number of chestnuts should be in the X-axis (independent variable). Please address accordingly.
L180-184: This section is good. It clearly states that the chestnuts used came from the control, which also matches with the description in the methodology.
Figure 1A: When describing the X-axis and Y-axis, please use a more descriptive statement. Saying just “nuts” and “larvae” is vague. Instead of “larvae” the authors could say “Total number of emerging larvae” or just “Number of emerging larvae”. For “nuts”, “number of nuts” would be more appropriate.
Figure 1B: This one is very clear. The dependent variable was located in the Y-axis, which makes it easy to understand. Good job!
L185-191: is this section the results for the microcosms experiment? Apparently, it is not. I had to go back to realize these are the results for the second cohort of chestnuts that were only treated with water (i.e., control). I got confused because the authors used the word “buckets of chestnuts” instead of 1L plastic containers. Please be consistent when using your own words. Please address accordingly.
L191: ever? Do the authors mean even?
Figure 2: Please avoid using the word bucket in the description since this word has been associated with the microcosms experiment. Add “,” before respectively.
L193-194: Please consider the following “Biological control treatments significantly explained the differences in C. sayi responses”
L194-195: Considering this is a factorial design. Was there a significant interaction between treatments and year in the orchard? If not, is should be mentioned. If the interaction was significant, it should be stated as well. In this case, the interaction (treatments*year) becomes more important than the main effects (treatments and year).
Figure 3D. I believe the authors forgot to include an asterisk (*) denoting which treatment was significant.
L206: Why suddenly the authors start using the standard deviation (SD) when the results show the Standard Error (SE). Both measure variability; however, they are calculated differently. Please correct.
L214-215: “a particular popular chestnut”. This is not a scientific term, and it should never be used in scientific publications. Please consider: “However, unexpectedly, one single chestnut contained 12 larvae.
L210-212: “While early reports of C. sayi emergence documented the ability for multiple larvae to emerge from a single nut” If there are reports, these should be cited. Please include references.
L218-220: “There were no obvious differences in the health of the larvae…” The authors did not measure if larvae were healthy or not, it was not part of the study. This is just an assumption and should be written as one. It is unknown if several larvae from emerging from a single nut can reach the adulthood stage.
L228-229: “a practice that involves waiting for two weeks (14 days) to determine which nuts are not infested does not seem adequate”. This statement needs a reference. Who says 14 d are enough to determine the infested nuts? Please provide reference.
L239: Please replace “bacteria” for “bacterium”. Bacteria is plural, bacterium is singular.
L243-244: Please consider: “Their presence could be a contributing factor to the continued decline of nut weight over time”.
L249: “compared to controls”. There was more than one control?
L251-252: “entomopathogenic fungi to the trunks of chestnut trees could have a served as a repellent to adult C. sayi”. I am intrigued about this statement. Why they authors think that ENF can be acting as repellent? I have not heard of this before. Could the authors provide references that show this as a possibility. Without references this is just speculation and has no scientific validation.
The second factor is more believable and there are several references that prove it.
L260: It is unlikely that EPF act as deterrent. I don’t believe they produce volatile organic compounds that can act as deterrents. Please rephrase.
L267-269: Just saying “previous work by our lab group” is not enough. This should be supported with references. If these references come from the same lab, include them, references should always be provided, specially with this kind of statements.
L271: “it is possible that the effect might be lagged”. I like this hypothesis. The authors should expand this idea more. For instance, why would it be late? How “fast” can the nematodes move in the soil? Do they have a hard time finding the larvae?
L277-280: These sentences are good. Good job!
Author Response
Thank you for your feedback. We appreciate your suggestions and detail how we have incorporated them in the accompanying attachment below.

Reviewer 2 Report
I have carried out review of the article “The Lesser Chestnut Weevil: Damage and Management with Biological Control” by Filgueiras and Willett.
I attest the study was well conducted, and quite interesting. In addition, the manuscript was well written and the organization is of good standard. I do not have any reservation against the acceptance of the manuscript for publication in the insect journal.
However, I suggest the authors might, perhaps, consider revising the title, as it appears too broad and with less details in its current expression. I assume it would attract more interest from readers if the title clearly indicates the study involves the application of entomopathogenic nematodes and entomopathogenic fungi in combination for the management of lesser chestnut weevil (Curcilio sayi)
L71 – The statement – ‘Both these options had been used for control of the related for C. elephas but never used’ is incorrect.
L157 – 158 – I assume the statement here – ‘Nut damage by emerging C. sayi larvae were visualized using histograms and density distributions and modeled using generalized additive models.’ – is missing a ‘,’.
L191 – ‘ever after’ should rather be ‘even after? ‘
L210 – ‘C. sayi’ as it appears at the start of the paragraph should have the genus name written in full.
L218 – The statement could be revised as ‘advantages or disadvantages’
L279 – Please, check this statement again. It appears it is missing a word.
Author Response
Thank you for your feedback and suggestions. We have incorporated them as detailed in the attachment below.

Reviewer 3 Report
The lesser chestnut weevil is an important pest of chestnut. In this study, the damage by the weevil was evaluated in a commercial orchard, and options for biological control, entomopathogenic fungi, entomopathogenic nematodes, and their combination were explored over two years. The research topic is original, results and findings are significant. I suggest the consideration for publication after minor revision.
Comment:
1. Introduction, add main biology characteristics of the lesser chestnut weevil.
2. Methods are not adequately described, and can be improved:
2.1 What is the experimental design for the management study in the commercial orchard? Number and size of plot should be described in the entomopathogenic nematode experiment.
2.2 How to apply the entomopathogenic fungi to the trunk? How many trees are used for 1 treatment/replication?
2.3 Line 116: What is the recommended rate of the entomopathogenic fungi?
2.4 Do the larvae damage both in the soil and inside the chestnut? I think the result should be checked also in the soil, not just the chestnut, as the entomopathogenic nematodes were sprayed in the soil.
2.5 It would be better to separate the 2 nematode species, not mix together.
2.6 Microcosms experiment, which instar larvae were used?
3. Describe the result from orchard and from microcosms experiment separately.
4. Discussion: Line 251-252, is there any reference about entomopathogenic fungi serves as a repellent to weevil adult?
Author Response

(The authors gave the same response as above.)

Round 2
Reviewer 1 Report
Thanks to the authors for addressing/explaining my comments/concerns.
The manuscript reads better. Please find below a few additional comments:
L5: add “,” after the word cases
L23: Since most of the keywords (3) were included in the title, please consider replacing them with other words that can increase the visibility of your study.
L42: delete “being”
L96: Please consider the following title: “Entomopathogenic nematode strain evaluation under lab conditions”.
L117-118: Please consider the word “obtained” or “collected” instead of “isolated”. Isolated from upstate NY does not seem appropriate.
L117-120: Please consider including the reason(s) of choosing these two species of nematodes in the methodology section. The reason is given in L298-299, but it would be more appropriate in the M&M.
L147: please replace “5” for “five”. Numbers less than 10 are used when a unit of the metric system is given (i.e., mm, cm, m, km, L, Kg, mg, etc.). Gallon plastic bucket is not from the universal metric system.
L147-149: Please consider:
“…into which 15-20 large diameter (about 7cm) holes were drilled at the sides and bottom of the bucket and then covered with a fine mesh screen (1000 micron)”
L214: Please consider the following title: “Entomopathogenic nematode strain evaluation under lab conditions”.
Figure 3. Interesting figure. It would be ideal to highlight the two species of nematodes used for the field study. I assume those are H. bacteriophora NY and S. feltiae NY. This would help the reader a lot.
L217-218: Please consider: “Exposure to nine nematode strains resulted in a significantly higher probability of mortality compared to their respective control”
Note of advice: The authors should mention that mortality probability was only compared against their respective control and not among strains.
L218: Please delete extra space in (Figure 3 )
L281-282: Please consider: “… could have potentially served as a repellent…”
L297: increased
L300-301: “In particular, one of the species we used (S. feltiae) can more than double mortality over controls”
The previous sentence does not make sense. Please, rephrase.
L: 301: delete “entirely”. Please avoid unnecessary words.
L304: Please consider “might be the result of” instead of “might be a result of”
Author Response
Thank you for your additional suggestions. We really appreciate the attention to detail. We have included your suggestions in the revised version. A detailed response is attached below.
